# Results of TETimaX Trial of Langerhans Cell Histiocytosis Treatment and Perspectives on the Role of Imatinib Mesylate in the Era of MAPK Signaling

**DOI:** 10.3390/biomedicines9121759

**Published:** 2021-11-24

**Authors:** Liliana Montella, Margaret Ottaviano, Vittorio Riccio, Fernanda Picozzi, Gaetano Facchini, Luigi Insabato, Mario Giuliano, Giovannella Palmieri

**Affiliations:** 1ASL NA2 NORD, Oncology Operative Unit, “Santa Maria delle Grazie” Hospital, 80078 Pozzuoli, Italy; gaetano.facchini@aslnapoli2nord.it; 2Oncology Unit, Ospedale del Mare, 80147 Naples, Italy; margaretottaviano@gmail.com; 3CRCTR Coordinating Rare Tumors Reference Center of Campania Region, 80131 Naples, Italy; m.giuliano@unina.it; 4Department of Clinical Medicine and Surgery, Università degli Studi di Napoli “Federico II”, 80131 Naples, Italy; vittorioriccio1990@gmail.com; 5Division of Medical Oncology, A.O.R.N. dei COLLI “Ospedali Monaldi-Cotugno-CTO”, 80131 Naples, Italy; fernanda.picozzi@ospedalideicolli.it; 6Anatomic Pathology Unit, Department of Advanced Biomedical Sciences, School of Medicine, University of Naples Federico II, 80131 Naples, Italy; luigi.insabato@unina.it

**Keywords:** langerhans cell histiocytosis, imatinib, B-RAF, KIT, PDGFR

## Abstract

Langerhans cell histiocytosis (LCH) is a rare disease that has a variable clinical presentation and unpredictable behavior. Until recently, therapeutic options were limited. Insights into the role of mitogen-activated protein kinase (MAPK) signaling have allowed the increased use of targeted treatments. Before the advent of drugs that interfere with this pathway, investigations concerning the tyrosine kinase inhibitor imatinib opened the way to a rationale-based therapeutic approach to the disease. Imatinib block the binding site of ATP in the BCR/ABL protein and is also a platelet-derived growth factor receptor (PDGFR) and a KIT (CD117) kinase inhibitor. A case of refractory LCH with brain involvement was reported to be successfully treated with imatinib. Thereafter, we further explored the role of this tyrosine kinase inhibitor. The present study is composed of an immunohistochemical evaluation of PDGFRβ expression and a clinical evaluation of imatinib in a series of LCH patients. In the first part, a series of 10 samples obtained from LCH patients was examined and a strong immunohistochemistry expression of PDGFRβ was found in 40% of the cases. In the clinical part of the study, five patients were enrolled. Long-lasting disease control was obtained. These results may suggest a potential role for this drug in the current age.

## 1. Introduction

Langerhans cell histiocytosis (LCH) is a rare disease (4.6 cases per 1 million among children under 15 years of age; 1–2 cases per million among adults) and its origin was long a matter of debate. LCH is characterized by the infiltration of tissue, such as skin, bone, brain, lung, and liver tissue, by the hallmark CD1a+ CD207+ cells, such as Langerhans cells (LCs), eosinophils, neutrophils, macrophages, and lymphocytes. The hallmark cells are present in variable percentages in LCH lesions, ranging from 1% to 70% [1]. The clonally expanded LCs suggest the presence of pathogenetic mechanisms such as those involved in hematological and cancerous disease. LCs belong to the family of dendritic cells that derive from hematopoietic medullary precursors. Recently, the detection of clonal mutations in genes of the mitogen-activated protein kinase (MAPK) pathway placed LCH near to myeloid disorders, given the presence of benign histiocytes [1]. Given these premises, LCH can be considered an atypical myeloproliferative disease. 

Systemic histiocytic neoplasms differentiate LCH from non-LCH diseases, including, in the latter group, Erdheim–Chester disease (ECD), juvenile xanthogranuloma, and Rosai–Dorfman disease. However, at least partially, they share some mutations. To reflect these overlaps, the most recent classification [2,3] identified and designated five groups: L (Langerhans), C (cutaneous and mucocutaneous), M (malignant), R (Rosai–Dorfman), and H (hemophagocytic). This classification is based on the origin of myeloid cells hit at different stages of differentiation and shared mutations in the MAPK pathway [4]. 

LCH is frequently associated with a paraneoplastic syndrome, such as diabetes insipidus, because of the infiltration of the hypophysis-surrounding tissue by LCH cells. The clinical features of the disease can be variable and their evolution over time is unpredictable. Multisystem LCH is frequently recognized in adults and children. LCH is a chronic disease and therefore can affect patients’ and their families’ quality of life. Therapeutic options are lacking and the recruitment of patients for research studies is difficult given the rarity of the disease. There are only clinical guidelines for children with LCH, which are extremely aggressive and require high-dose chemotherapy and/or bone marrow transplantation.

Imatinib mesylate is a small molecule analog of adenosine triphosphate with 2-phenylaminopyrimidine as an active part, and was created specifically to block the binding site of ATP in the BCR/ABL protein. It is also a platelet-derived growth factor receptor (PDGFR) and a KIT (CD117) kinase inhibitor. The PDGFR family consists of two receptors, an alpha and a beta-receptor, that signal through the cell surface of the tyrosine kinase receptors (PDGFR) and stimulate various cellular functions including growth, proliferation, and differentiation [5]. Its kinase-inhibiting activity makes this molecule the ideal candidate for the treatment of BCR/ABL+ chronic myeloid leukemia and CD117+ gastrointestinal stromal tumors. As in the “key in the lock” model, finding one of these driver mutations in tumors can address the use of imatinib mesylate. KIT corresponds to a 145-KD transmembrane glycoprotein and is a member of the receptor tyrosine kinase closely related to the receptors for PDGF, the macrophage colony-stimulating factor, and the FMS-like receptor tyrosine kinase (FLT3) ligand [6]. The KIT expression has been documented in acute myelogenous leukemia and mastocytosis [6]. Mutations of KIT are mostly found in cells such as mast cells, germ cells, and hematopoietic stem cells whose development depends on the Steel factor (SLF)/KIT axis activation. Early investigations showed a relationship between PDGF and monocytes. Increased production of PDGF by alveolar macrophages was shown in lung LCH [7]. Moreover, stimulated monocytes were shown to express c-sis proto-oncogene, which encodes one of the PDGF chains [8]. A combined immunohistochemical and molecular cytogenetic investigation of S100, CD1a, c-KIT, and PDGFRα and β showed that a discrete percentage (approximately 30% of 14 studied cases) showed expression of PDGFRα [9].

PDGFR-β and other proteins, such as p-Akt (Ser-473), p-mTOR (Ser-2448), and p-p70S6K (Thr-389), that are part of PDGFR-β signaling, contribute to an integrated circuit that fuels histiocyte growth and apoptosis resistance [10].

Moreover, preclinical, and clinical data support a role for imatinib in inducing not only direct on-target, but also indirect off-target effects.

During imatinib treatment, through c-KIT signaling, NK cells are activated, a favorite production of Th1 over Th2 cytokines is induced, and tolerogenic mechanisms are suppressed [11,12]. As shown in leukemia, imatinib influences normal hematopoietic cells by inhibiting the differentiation and functioning of dendritic cells, and, as shown in in GIST, imatinib improves NK response, boosts NK cell-induced IFNα secretion, and decreases regulatory T-cells in patients [13]. 

Imatinib mesylate was shown to be effective in PDGFRα/β positive LCH with brain involvement [14] and in CD117+ thymic carcinoma [15]. Given these premises, we evaluated PDGF-Rβ expression by immunohistochemistry in a series of 10 LCH cases [16]. Moreover, a phase 2 study was started to define the activity of imatinib in a series of patients. 

In this report, the immunohistochemical evaluation of KIT/PDGFRβ on 10 LCH samples and the updated clinical results with imatinib in a separate series of LCH patients are described.

## 2. Materials and Methods

### 2.1. Immunohistochemistry Study

Ten biopsies (7 males/3 females, age range: 1–76 years) were evaluated. Two biopsies of oral mucosa, performed at different times, were available for one patient.

The tissues available were bone in 4 patients, skin in 2 patients, and 1 case each of oral mucosa, brain, lymph node, and ocular soft tissue. For the bone tissue, the fragments were processed without decalcification. A 4 μm section obtained from the formalin-fixed, paraffin-embedded specimens was incubated in a microwave oven for 15 min in 10 mmol/L, 6.0 pH buffered citrate following the immunohistochemical procedure for platelet-derived growth factor receptor beta (PDGFRβ) (rabbit polyclonal ab, Santa Cruz Biotechnology Inc., Santa Cruz, CA, USA) and KIT (CD117) (Dako, Carpinteria, CA, USA), diluted to 1:50 and 1:300, respectively. The conventional avidin–biotin complex procedure was applied according to the manufacturer’s protocol. The section was briefly washed and incubated with a primary antibody overnight at 4 °C, and then incubated with a secondary antibody. Positive staining was revealed using DAB chromogen, according to the supplier’s conditions, followed by counterstaining with Mayer Hematoxylin. The slide was cover-slipped with a xylene-based mounting medium. The diagnosis of LCH was based on the presence of a population of Langerhans cells showing an unequivocal positive reaction to CD1a and S-100 protein antibodies mixed with eosinophils and inflammatory cells. The immunostaining with PDGFRβ was scored from 1 to 3, with 1 corresponding to low expression and 3 to strong expression of the Langerhans cells in the cytoplasm. 

### 2.2. TETimaX Trial

The primary objective of the study was the evaluation of drug activity based on the response evaluated by RECIST criteria. Secondary objectives were the evaluation of safety and effects on quality of life. Safety was assessed according to National Cancer Institute criteria. Twenty patients aged between 18 and 75 years old, with WHO performance status 0–1 and histological diagnosis of LCH were expected to be enrolled. The study protocol was approved by the Local Ethical Committee and was submitted to Agenzia Italiana del Farmaco (AIFA) (TETimaX trial, EUDRACT NUMBER: 2007-006119-22 AIFA CODE: FARM6HJ7CA). The TETimaX trial enrolled patients with refractory thymic epithelial tumors and LCH. Informed consent was required and obtained before study enrollment. Patients had received previous treatment and showed disease progression. Immunohistochemical evaluation of cKIT, PDGFRα, PDGFRβ, mTOR, or p-AKT was contemplated. At least one, measuring either ≥2 cm or ≥1 cm if identified by computed tomography (CT) scan, was necessary. Patients were required to have normal laboratory exams to assess bone marrow, renal, and liver function. Before study entry, patients were evaluated with appropriate scans such as X-rays, CT, bone scintigraphy with ^99m^Tc-methylene diphosphonate, nuclear magnetic resonance (NMR), positron emission tomography (PET), and ortopantomography. Exclusion criteria were pregnancy, breastfeeding, absence of adequate contraception, unmeasurable lesions as the only site of disease (for example, pleural effusion, ascites, peritoneal carcinosis, bone metastases), uncontrolled heart disease (arrhythmias, unstable angina), or thromboembolism sequels (heart attack, stroke, transient ischemic attack, embolism, or venous thrombosis), other uncontrolled diseases, chemotherapy in the 30 days before study enrollment, or inability to express informed consent and to follow study protocol. 

Imatinib mesylate was administered at a daily dosage of 400 mg for one year if an objective response was documented. Drugs interfering with isoenzyme CYP450 (2D6 and 3A4) can reduce the activity of imatinib, therefore they were allowed with caution. Furthermore, patients taking concomitant drugs metabolized by P-450 cytochrome underwent more follow-up visits because of the increased risk of toxicity. Concomitant use of acetaminophen was not allowed. Warfarin was not allowed because of P-450 cytochrome metabolism and was substituted with low-molecular-weight heparins. Time to tumor progression (TTP), defined as the time elapsed between treatment initiation and disease progression, was assessed. Response assessment by the same radiologic technique used at basal evaluation was performed every three months. Safety was evaluated according to NCI CTCAE v3.0 criteria and classified into grade 1: low risk, 2: moderate risk, 3: severe risk, and 4: life-threatening. Colony-stimulating factors (CSF) were allowed per protocol in the case of grade 4 febrile neutropenia, grade 3 and 4 neutropenia, and grade 4 neutropenia associated with diarrhea. In these instances, prophylactic use of CSFs was allowed in the following cycles. Radiotherapy and surgery could be used during the study if medically required.

## 3. Results

### 3.1. Immunohistochemistry Results

A comprehensive picture of the results obtained is shown in Table 1. A strong immunohistochemical expression of PDGF-Rβ was found in four cases. In Figure 1, a case of strong cytoplasmic expression of PDGFRβ is represented. 

Tissue differential expression was not detected among the cases that included bone or oral mucosa. Four cases received an intermediate score for cytoplasmic expression and one case displayed a low cytoplasmic expression of PDGF-Rβ. The immunohistochemistry series also included the brain biopsy of the patient previously described and treated with imatinib mesylate [14]. We evaluated two tissue samples from oral mucosa obtained at different times from the same patient. A strong expression of PDGF-Rβ was detected in both specimens. A low expression of PDGF-Rβ was revealed in a patient with ocular soft tissue infiltration by LCH. KIT returnednegative results in all cases of this series.

### 3.2. TETimaX Trial Results

The protocol was stopped in 2013 because of low accrual. From 2008 to 2011, five patients (three males/two females, age range: 25–60 years, median 41 years), followed in a single Center, were enrolled. Given the small number of patients enrolled, as compared to expected, all patients were considered assessable for drug activity and efficacy. In Table 2, patients’ characteristics, responses, and current state are reported.

Unfortunately, immunohistochemical evaluation of the patients enrolled in the clinical study was not performed because of a lack of available materials. The absence of the B-RAF mutation was only defined for one patient. This evaluation was performed through the amplification of a segment within exon 15 of BRAF and gene sequencing by Sanger.

All patients were pretreated, as shown in Table 2. The patients had received 6-month treatments with vinblastine plus prednisone, cladribine, and in one case indomethacin as experimental treatments. Three patients had lung LCH localizations and one patient had eye involvement. All patients had diabetes insipidus. One patient had multiple pituitary dysfunctions. One patient had one site of LCH involvement, while the others had multisystem LCH. Three patients had multi-bone involvement. Among the patients with lung involvement, one patient had pituitary and brain LCH. One patient had retroperitoneal fibrosis, which is suspected to be related to the LCH infiltrative process. However, no informative material obtained by a biopsy was available. Four out of five of the patients completed one year of treatment. One patient stopped imatinib after 4 months because of atrial fibrillation (Patient n.4). As concerns the response, we registered one complete response, three stable diseases, and one mixed response, i.e., a partial response in the brain LCH and stable disease in the lung LCH. The response was confirmed at the second radiological evaluation. Moreover, one patient with lung and bone LCH involvement reported a complete response at the last evaluation (Patient n.1, Figure 2); thus, he averted the expected heart–lung transplantation. 

The TTP was variable, ranging from 1 to 120 months (median 108 months). Three patients were free of treatment at the date of the last evaluation. The patient (n.3) who was pretreated with two lines of chemotherapy before study enrollment reported a complete brain and bone response and a partial response for lung disease as the best response and remained free of further treatment at the date of the last evaluation. Two patients received further treatment after imatinib. One of these was the patient that prematurely stopped imatinib (patient n.4). He received interferon for two years and three months, then relapsed and was treated with vemurafenib for two years and two months based on compassionate use. He was free of disease; however, he died because of end-stage renal disease, secondary to obstructive uropathy from retroperitoneal fibrosis. The other patient who had bone-only disease (Patient n.5) relapsed after one year and received vinblastine plus prednisone and etoposide following the relapse. All but one patient (Patient n.4) were alive at the date of last contact (July 2021). Treatment was well-tolerated, with mainly 1–2 graded nausea related to imatinib reported.

## 4. Discussion

Treatment of LCH encompasses several options, ranging from chemotherapeutics to immune-modulatory and targeted drugs. When evaluating activity and efficacy, clinicians should keep in mind the unpredictable behavior of the disease. Pediatric and adult LCH appear to be different in clinical features and response to treatments. 

Following the report of the successful treatment of a patient with neurological refractory LCH with imatinib, interest in imatinib use increased. The activity was shown in three patients without significant genomic correlation, because of a lack of materials and/or KIT/PDGFRA mutations [17]. Uncertain impressions can be drawn from the examination of available reports. They are mostly single-case reports. One pretreated patient with Rosai–Dorfman disease showed a complete response to imatinib [18]. In other reported cases, the treatment with imatinib failed [19].

We found a significant expression of PDGFRβ in LCH. All biopsies in the present study returned positive results. This finding encouraged us to explore the activity of imatinib in a series of LCH patients. 

A previously published series documented PDGFR expression in LCH. A positive PDGFR-β expression in histiocytes was found in 86.5% (32 of 37 samples) of ECD and 27.3% (3 of 11) of LCH biopsies [20]. Nevertheless, the response to imatinib was not clearly assessable. A formal comparison between our immunohistochemistry results and those reported by Haroche et al. cannot be performed because of the limited number of cases studied and the small sampling available. 

Despite the plan for 20 patients to be enrolled in the TETimaX trial, only five patients were recruited. All but one patient had multisystem involvement. Imatinib was shown to control the disease in all patients. A response was also achieved in the pretreated case. At a prolonged follow-up –more than a decade later three patients were free of treatment and all but one were alive. 

This study has several acknowledged limitations. The trial was originally designed to be multicentric, with 20 patients predicted to be enrolled. However, the study was prematurely stopped because of the low accrual and because it was only monocentric. Therefore, the limited number of patients enrolled and the lack of a control group reduce the relevance of our findings, as is easily acknowledged. However, in LCH more than in other diseases, it is particularly difficult to plan such studies. Unfortunately, none of the patients in the immunohistochemistry series were enrolled in the clinical part of the study. The critical limitation, while quite common, especially in retrospective series, was the lack of tissue from which to evaluate genomic alterations. In the clinical series, the biopsies available for pathological diagnosis included lung, eye, and temporal bone tissue. Biopsies from these sites are often only sufficient for diagnosis, and repeated sampling is not easily accepted by the patients. This issue can be overcome with liquid biopsies in the future. Moreover, we highlight that immunohistochemistry is limited to protein detection, and genomic studies to detect molecular aberrations are also needed. In addition, the choice to treat patients with an experimental drug for a definite, instead of indefinite, time (one year) may be disputable. We note that one of the two patients who relapsed stopped the imatinib treatement prematurely because of atrial fibrillation. Moreover, in this series, the choice of a 1 year treatment can be considered appropriate because the control of the disease was maintained in the responsive patients and they were protected from the toxicities related to longer-term therapies. This kind of treatment may be particularly convenient for LCH patients because of the unpredictable disease behaviour with flares alternating with dormancy that further complicated management and therapy. However, LCH may severely worsen quality of life because of the multiplicity of systemic symptoms and long-term morbidity. Despite the previously cited limitations, this study reintroduces the need for accurate basal genomic assessments of LCH and cooperative multicentric studies which can overcome the recruitment difficulties. 

The increasing efforts to define pathogenetic mechanisms and identify targeted drugs have focused attention on MAPK activation. A breakthrough in understanding the pathogenesis of LCH occurred in 2010 when a gain-of-function mutation in B-RAF (V600E) was identified in more than half of LCH patient samples [21]. Similar to in melanoma [22], colon [23], and thyroid cancer [24], BRAF V600E marks poor outcomes. In a series of 315 children with LCH, the B-RAF V600E mutation was correlated with aggressive disease and resistance to chemotherapy [25]. 

MEK, also known as mitogen-activated protein kinase MAP2K1, and ERK phosphorylation were also found in 100% of examined cases [26]. The ERK pathway is activated in all cases, including those with wild-type B-RAF alleles. MAP2K1 mutations have been reported in BRAF-negative cases. ERK was resistant to B-RAF-V600E inhibition but responsive to both a second-generation BRAF inhibitor and a MEK inhibitor [27]. 

The discovery of additional mutations in LCH including A-RAF, N/KRAS, PIK3CA, and gene fusions involving B-RAF, ALK, and NTRK [28] enrich the variegate landscape of the histiocytosis and further complicate the therapeutic choice. 

The identification of the specific MAPK pathway mutation present in each patient with LCH could enable personalized approaches to medication and the design of treatment sequences. In line with this logic, the Pediatric MATCH Screening trial (NCT03155620) [29] imposes treatments based on genetic testing results for solid tumors, as well as histiocytic disorders. 

The present study started before the identification of the role of the BRAF mutation in LCH. In our small series of LCH patients, imatinib mesylate was able to produce a significant clinical benefit rate and prolonged progression-free survival. Imatinib treatment was also found to produce an objective response in a patient with brain and bone LCH involvement previously treated with two lines of therapy. It is important to note that one patient from the present series is not evaluable for progression-free survival as he stopped the treatment prematurely.

Imatinib blocked multiple downstream signaling pathways activated by cKIT, PDGFRA, and bcr-abl, including those mediated by phosphatidylinositol 3-kinase (PI3K)/Akt, Ras/RAF/MEK/ERK, and the signal transducer and activator of transcription (STAT), which are involved in cell proliferation. We suppose that the effect of imatinib is also mediated through interference with the MAPK pathway, which is frequently involved in LCH. 

The role of c-KIT in comparison to BRAF mutations in LCH is not well defined. In one recent study, c-KIT and MAP2K1 mutations were evaluated in skin pathological samples of patients with cutaneous LCH labeled as BRAF wild-type [30]. MAP2K1 is the gene that encodes MEK. The c-KIT and MAP2K1 mutations were represented at similar percentages (57% and 43% of LCH cases, respectively). We note the role of KIT inhibitors as carriers of such mutations. Moreover, the simultaneous presence of the c-KIT and MAP2K1 mutations suggest a role for the dual inhibition of KIT and MEK. We can suppose the following algorithm for LCH: targeted treatments when the driver mutations have been identified, and imatinib when the target mutations are undetectable and at the first line failure. At present, the unknown pathogenesis of about half of the cases classified in the R-group of LCH also translate into unmet medical needs. 

In conclusion, the crucial role of BRAF inhibition does not definitively exclude the role of imatinib in LCH. We can suppose the therapeutic relevance of the presence of cKIT mutations and/or synergistic activity together with other agents interfering with MAPK signaling. Larger studies, possibly including the identification of individual driver mutations, could contribute to defining the best-tailored approach. 

## Figures and Tables

**Figure 1 biomedicines-09-01759-f001:**
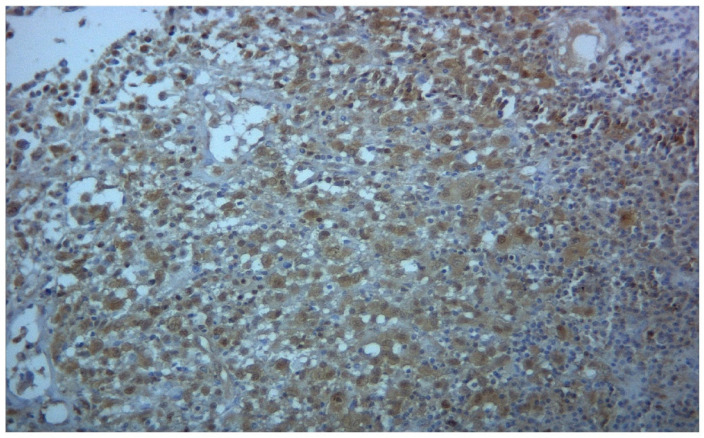
Immunohistochemistry for PDGFRβ shows strong cytoplasmic positivity in Langerhans cells (magnification 106×).

**Figure 2 biomedicines-09-01759-f002:**
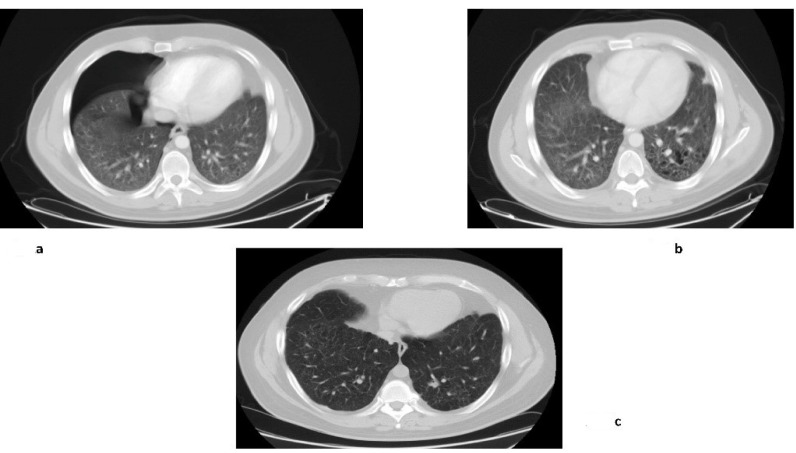
In these pictures, the response obtained with imatinib in Patient n.1 is reported: (**a**) Computed tomography of the chest shows the basal condition of the patient with pneumothorax; (**b**) Improvement after one year of treatment; (**c**) Maintained response ten years later.

**Table 1 biomedicines-09-01759-t001:** Immunohistochemical evaluation of PDGF-Rβ in 10 patients. Strong expression of PDGFRβ was outlined by bold character. C: cytoplasmic; N: nuclear.

Case	Sex/Age	LCH Site	PDGF-Rβ
1	M/10	Ocular soft tissue	+1C
2	F/9	Skin	**+3C/N**
**3**	M/76	Skin	+2C
4	M/13	Bone	**+3C**
5	M/19	Bone	**+3C**
6	M/1	Bone	+1C
7	M/45	Oral mucosa	**+3C**
	M/45	Oral mucosa	**+3C**
8	F/23	Brain	+2C
9	M/5	Eye	+2
10	F/7	Bone	+2

**Table 2 biomedicines-09-01759-t002:** CR: complete response; PR: partial response; SD: stable disease; A: alive; D: dead; yrs: years; vem: vemurafenib; VP16: etoposide; BS: ^99m^Tc bone scintigraphy; CNS: central nervous system; CT: computed tomography; CUT: cutaneous involvement; CV: cardiovascular system; F: female; GI: gastrointestinal tract; HTN: hypertension; IFNa: interferon alpha; pIFNa: pegylated interferon alpha; LYM: lymph nodes; m: month; M: male; MNG: meninges; MRI: magnetic resonance imaging; MTX: methotrexate; PET/CT: positron emission tomography/computed tomography; PIT: pituitary gland; PLM: pulmonary system; PDN: prednisone; ROS: retro-orbital space; RP: retroperitoneum; RTN: retina; STN: skeleton; UNK: unknown; Vbl: vinblastine; VMR: vemurafenib; w: weeks; wt: wild type.

Pt	Age	Sex	Presenting Sign/Symptom	Disease Involvement Sites	Best Responses and Timeframe	Treatments before Imatinib	Treatments after Imatinib	Follow-up after Imatinib	Current State Notes
1	25	M	Cough	Lung, multi-bone	CR Heart–lung transplantation avoided (10 yrs)	Vbl + Pdn	-	12 years + 1 month	A
2	46	F	Cough, weight loss	Lung	PR (10 yrs)	Vbl + Pdn	-	11 years + 6 months	A
3	41	F	Cough, headache, memory disturbances	Lung, brain, multi-bone	Brain and bone CR/lung PR (9 yrs)	Vbl + Pdn Cladribine Indomethacin	Cladribine Vbl + Pdn	8 years + 6 months	A
4	60	M	Diplopia	Bone, retroperitoneal fibrosis	SD	Vbl + Pdn	IFN→vem	9 years	D
5	36	M	Palpebral ptosis	Multi-bone	CR	Vbl + Pdn	Vbl + Pdn→VP16	12 years + 7 months	A b-raf wt

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
