# Peer review of "Results of TETimaX Trial of Langerhans Cell Histiocytosis Treatment and Perspectives on the Role of Imatinib Mesylate in the Era of MAPK Signaling"

_biomedicines, 2021, doi:10.3390/biomedicines9121759_

Round 1
Reviewer 1 Report
I sugest adding detailed results of the studies that support the role of Imatinib in Langerhans cell histiocytosis.
Author Response
Dear Reviewer,
Thank you for your advice. In the discussion, on page 7 lines 252 and following, we added some considerations. However, we highlight that few studies, mostly single reports, are available. Given the lacking evidence supporting imatinib in LCH, our experience could result significant to the readers involved in LCH treatment, despite recognized limits.
Kind Regards
Reviewer 2 Report
Very interesting case study, showing the probable benefits of imitanib mesylate in the treatment of Langerhans cell histiocytosis. Authors are describing the PDGFRb expression in a cohort of 10 biopsies (9 patients) of LCH, and describe the response to imitanib mesylate treatment in a cohort of 5 separate patients. The potential benefit of Imatinib mesylate as adjunct therapy in LCH was discussed.
- Regarding the cohort used for PDGFR immunohistochemical expression:
It is better to specify at the beginning of the material and method paragraph (line 105), that two of the 10 biopsies corresponded to the same patient and were performed at different times in the course of the disease. (Maybe it’s better to start with “ten biopsies” instead of “ten cases”)
Some of the specimens examined included bone tissue. If available, please add details about the tissue decalcification process (if applicable) before embedding (for example, was a regular decal used or EDTA based… etc)
Line 172: Authors mention that one biopsy was negative for PDGFRb expression: This does not seem to be reflected in the table #1, which shows that 10 biopsies were positive with variable intensity.
Was the patient with 2 biopsies treated in between the 2 biopsies?
In the discussion paragraph, line 241: Please clarify that in the cohort of 10 biopsies from 9 LCH patients, we found a significant immunohistochemical expression of PDGFRb (90%? 100%? Add percentage). Is there any explanation for the difference form literature results (27% per Haroche et al)? or is it simply due to the small sampling?
- Regarding the cohort of the treated patients:
Line 133: Authors mention that patients had received a previous treatment and showed disease progression. If possible, please give more details about the clinical course and type and duration of the treatment they had received prior their enrolment in this study
Table 2:
- If possible please split the treatment column in two, prior to and following Imitanib mesylate therapy.
- Please add an additional column for the duration of follow up after initiating imitanib mesylate therapy.
In patients with lung involvement, and in particular patient 4 who has only pulmonary manifestations, please comment on the smoking status along the course of the disease, given the strong association of smoking and pulmonary LCH. Was any of the patients smoker? Did they stop smoking after diagnosis? etc...
Line 199: Was the absence of BRAF mutation established by molecular testing or by negative immunohistochemistry?
Line 206: Authors mention that retroperitoneal fibrosis RPF was due to the infiltrative process by LCH. How was this determined? Was there a biopsy of the retroperitoneal process showing Langerhans cell infiltrate? If not, it may be better to just mention the RPF without attributing it to an infiltration by LC. RPF could be associated with LCH in this patient, without necessarily retroperitoneal involvement by Langerhans cells. For instance, retroperitoneal fibrosis can sometimes be seen in mixed histiocytoses, where the histiocytes in the peritoneum are non-Langerhans cells (Blood (2014) 124 (7): 1119–1126.).
- General:
Discussion line 310-311: Please specify percentages. ie: ckit and MAP2K1 mutations were found in 57% and 43% of LCH cases.
Discussion: Study limitations: please discuss also the low number of cases and absence of control group
Please mention or discuss literature cases of LCH that failed Imatinib treatment (example: Arch Dermatol. 2009;145(8):949-950. doi:10.1001/archdermatol.2009.164 )
Conclusion (lines 318-322): please clarify the conclusion, maybe different wording could help to make it stronger.
Author Response
Dear Reviewer,
Thank you for the favorable opinion expressed. Following, you will find a point by point reply to your observations:
- Regarding the cohort used for PDGFR immunohistochemical expression:
It is better to specify at the beginning of the material and method paragraph (line 105), that two of the 10 biopsies corresponded to the same patient and were performed at different times in the course of the disease. (Maybe it’s better to start with “ten biopsies” instead of “ten cases”)
Modified accordingly
- Some of the specimens examined included bone tissue. If available, please add details about the tissue decalcification process (if applicable) before embedding (for example, was a regular decal used or EDTA based… etc)
On page 3, line 108, we specify: Some scant bony fragments were processed without a decalcification process.
- Line 172: Authors mention that one biopsy was negative for PDGFRb expression: This does not seem to be reflected in table #1, which shows that 10 biopsies were positive with variable intensity.
On Page 4 Line 175: one case negative is wrongly reported.
- Was the patient with 2 biopsies treated in between the 2 biopsies?
Yes, the patient with two biopsies was treated with vinblastine plus steroids.
- In the discussion paragraph, line 241: Please clarify that in the cohort of 10 biopsies from 9 LCH patients, we found a significant immunohistochemical expression of PDGFRb (90%? 100%? Add percentage). Is there any explanation for the difference form literature results (27% per Haroche et al)? or is it simply due to the small sampling?
Page 7 line 262: A formal comparison between our immunohistochemistry results and those reported by Haroche et al. cannot be performed because of the limited number of cases studied and small sampling available.
- Regarding the cohort of the treated patients:
Line 133: Authors mention that patients had received a previous treatment and showed disease progression. If possible, please give more details about the clinical course and type and duration of the treatment they had received prior their enrolment in this study
On page 6 line 240: The patients had received 6-month treatment with vinblastine plus prednisone, cladribine and in one case indomethacin as experimental treatments.
- If possible please split the treatment column in two, prior to and following Imitanib mesylate therapy.Please add an additional column for the duration of follow up after initiating imitanib mesylate therapy.
Table 2 has been modified accordingly.
- In patients with lung involvement, and in particular patient 4 who has only pulmonary manifestations, please comment on the smoking status along the course of the disease, given the strong association of smoking and pulmonary LCH. Was any of the patients smoker? Did they stop smoking after diagnosis? etc...
All the patients were no smokers.
- Line 199: Was the absence of BRAF mutation established by molecular testing or by negative immunohistochemistry?
On page 6, line 204: BRAF mutation was assessed by Molecular testing
- Line 206: Authors mention that retroperitoneal fibrosis RPF was due to the infiltrative process by LCH. How was this determined? Was there a biopsy of the retroperitoneal process showing Langerhans cell infiltrate? If not, it may be better to just mention the RPF without attributing it to an infiltration by LC. RPF could be associated with LCH in this patient, without necessarily retroperitoneal involvement by Langerhans cells. For instance, retroperitoneal fibrosis can sometimes be seen in mixed histiocytoses, where the histiocytes in the peritoneum are non-Langerhans cells (Blood (2014) 124 (7): 1119–1126.).
On page 6, line 213 we clarify that no pathological information was available as concerns retroperitoneal fibrosis.
- General: Discussion line 310-311: Please specify percentages. ie: ckit and MAP2K1 mutations were found in 57% and 43% of LCH cases.
On Page 8 line 341 : The percentages were reported.
- Discussion: Study limitations: please discuss also the low number of cases and absence of control group
On page 7 line 279: Therefore, the limited number of patients enrolled, and the lack of control group reduce the relevance of our findings, as is easily acknowledged. However, in LCH more than in other diseases, it is particularly difficult to plan such studies.
- Please mention or discuss literature cases of LCH that failed Imatinib treatment (example: Arch Dermatol. 2009;145(8):949-950. doi:10.1001/archdermatol.2009.164 )
On page 7, line 260: Ref.18
- Conclusion (lines 318-322): please clarify the conclusion, maybe different wording could help to make it stronger.
On age 8: The conclusion has been re-phrased as follows: We can suppose the following algorithm in LCH: targeted treatments when driver mutations have been identified, imatinib when the target mutations are undetectable and at the first line failure.
Best Regards
Round 2
Reviewer 1 Report
I consider that the article can be accepted in the present form